# Innovative Bicultured Lactic–Acetic Acid Co-fermentation Improves Jujube Puree's Functionality and Volatile Compounds

**Turkson Antwi Boasiako** [1], **Yuqing Xiong** [1], **Isaac Duah Boateng** [2], **Jeffrey Appiagyei** [2], **Yanshu Li** [1], **Kerry Clark** [2], **Afusat Yinka Aregbe** [1], **Sanabil Yaqoob** [1] **and Yongkun Ma** [1,*]

[1]    School of Food and Biological Engineering, Jiangsu University, Zhenjiang 202013, China;
       turk.sky92@gmail.com (T.A.B.); 13207917516@163.com (Y.X.); 17361011612@163.com (Y.L.);
       adesimpa2021@gmail.com (A.Y.A.); sanabil.yaqoob@ucp.edu.pk (S.Y.)
[2]    College of Agriculture, Food and Natural Resources, University of Missouri, Columbia, MO 65201, USA;
       boatengisaacduah@gmail.com (I.D.B.); boakyejeff@gmail.com (J.A.); clarkk@missouri.edu (K.C.)
*    Correspondence: mayongkun@ujs.edu.cn

**Abstract:** Jujubes (*Ziziphus jujuba* Mill), characterized by a rich profile of bioactive compounds, have been historically less exploited due to their unappealing sensory characteristics when dried, including delayed bitterness and a limited shelf life when fresh. Co-fermented jujube puree has emerged as a strategy for enhancing its functional food potential. This study examined the impact of 8-day bicultured Junzao jujube puree, employing both commercial and indigenous Chinese lactic and acetic acid bacteria. Our investigation encompassed an assessment of functionality (cellular profile, antioxidant properties, color, free amino acids, phenolic profiling, volatiles elucidation using headspace-solid phase microextraction gas chromatography–mass spectrometry (HS-SPME-GC-MS), aroma analysis using electronic nose), and microstructural analysis using scanning electron microscopy (SEM). Viable counts of bicultured purees showed probiotic effects exceeding 6–7 log CFU/mL. Strong positive correlations were observed between phenolic compounds (chlorogenic acid, rutin, *p*-coumaric acid) and antioxidant capacities (ABTS-RSA and DPPH-RSA). The darker color of raw jujube puree was modified, exemplified by a significant ($p < 0.05$) negative correlation between overall color difference and cyanidin 3-O-rutinoside ($R^2 = -0.768$). Purees, particularly those containing bicultures of *Lactobacillus helveticus* Lh 43 and *Acetobacter pasteurianus* Ap-As.1.41 HuNiang 1.01 exhibited the highest potential free amino acid content ($157.17 \pm 1.12$ mg/100 g FW) compared to the control ($184.03 \pm 1.16$ mg/100 g FW) with a distinctive formation of L-methionine in biculture of *Lactiplantibacillus plantarum* Lp 28 and *A. pasteurianus* Ap-As.1.41 HuNiang 1.01. The phenolic profile of *Lacticaseibacillus casei* Lc 122 and *A. pasteurianus* Ap-As.1.41 HuNiang 1.01 increased by 22.79% above the control (48.34 mg/100 g FW) while biculture: *L. helveticus* Lh 43 and *A. pasteurianus* Ap-As.1.41 HuNiang 1.01 were enhanced by 4.37%, with the lowest profile in *Lp. plantarum* Lp 28 and *A. pasteurianus* Ap-As.1.41 HuNiang 1.01 (46.85 mg/100 g FW). The electronic nose revealed the predominant presence of sulfur, terpenes, and alcohol sensor bioactives in the fermented purees. HS-SPME-GC-MS analysis identified 80 volatile compounds in the bicultured purees, with esters constituting the major group (42%). Furthermore, SEM analysis unveiled massive microscopic alterations in the bicultured purees compared to the unfermented puree. These results collectively demonstrated that lactic–acetic acid co-fermentation serves to biovalorize Junzao jujube puree, enhancing its organoleptic appeal and extending its shelf life.

**Keywords:** jujube; lactic and acetic acid bacteria; free amino acid; SEM; phenolic compounds; HS-SPME-GC-MS

## 1. Introduction

*Ziziphus jujuba* Mill, commonly known as jujube, is a fructiferous tree from the *Rhamnaceae* family [1] and is extensively cultivated for its sweet and nutritive drupes [2]. The

historical use of the jujube tree in traditional medicine and gastronomy emphasizes its cultural significance [2]. Renowned for its adaptability to diverse climates, the jujube produces small, date-like fruits, typically red, with a unique blend of sweetness and chewiness [1]. Apart from its palatability, scientific interest in the jujube has grown due to perceived health benefits linked to its nutritional composition (rich in sugars, acids, pectin, and minerals) and bioactive constituents encompassing vitamins, minerals, fibers, amino acids, polysaccharides, polyphenols, flavonols, phenolic acids [1]. Despite its versatility and cultural importance, the exploitation of jujube's benefits is hindered by limited organoleptic appeal, delayed bitterness [3], and a short shelf life resulting from high moisture and sugar content [4]. Consequently, there is a pressing need for technologies that enhance jujube fruit's shelf life, nutritional quality, organoleptic characteristics, and health benefits, fostering increased commercial production for economic and health gains.

Fermentation, a deeply ingrained culinary practice, showcases a nuanced synergy of biochemical and microbiological processes [5]. Among the oldest and most economical biotechnological methods, lactic acid fermentation is valuable in enhancing fruits' nutritional, sensory, safety, and shelf-life [6]. Acetic acid bacteria (AAB) are equally essential in fermentation, transforming substrates into valuable products by oxidizing ethanol to acetic acid, influencing organoleptic qualities, safety, and shelf life [7]. The symbiotic relationship between lactic acid bacteria (LAB) and acetic acid bacteria (AAB) is crucial in influencing flavor, safety, and preservation in fermented foods and beverages, requiring a thorough understanding of their roles and synergies for fine-tuning fermentation processes [8].

The dynamic interactions between key components, including free amino acids, organic acids, phenolics, volatiles, and microstructure, are central to the transformative nature of fermentation. This intricate reciprocity influences flavor development [7], aroma [9], and texture in fermented foods. The collaborative actions of a diverse microbial community, such as bacteria, yeasts, and molds during fermentation [8], lead to the enzymatic breakdown of proteins, yielding free amino acids that are crucial as precursors to flavor compounds. Simultaneously, inherent or synthesized phenolic compounds during fermentation contribute to sensory attributes and provide antioxidant properties to the final product. Various phenolic compounds, such as quercetin 3-O-rutinoside, cyanidin 3-O-glucoside, pelargonidin 3-O-rutinoside, vanillic acid, caffeic acid, ferulic acid, *p*-coumaric acid, chlorogenic acid, *p*-hydroxybenzoic acid, and protocatechuic acid, have been identified in significant quantities in jujube [10–12]. Jujube is rich in phenolic compounds, contributing to sensory attributes like astringency and flavor. Beyond their sensory impact, these compounds offer various health benefits, including neuro-protective, cardio-protective, anti-inflammatory, antioxidant, cancer chemo-preventive, immunomodulatory, and antipyretic properties [1,10,11]. Their multifaceted roles as proton donors, metal chelators, reducing agents, and singlet oxygen quenchers underline their significance [13]. Furthermore, the antioxidant properties of jujube fruits enhance their sensory appeal and emphasize their potential as naturally health-beneficial compounds. The volatile compounds responsible for the characteristic aroma and flavor in fermented foods arise from the metabolic activities of microorganisms and the transformation of substrates, such as proteins [14] and lipids [15].

Microstructure, often overlooked in fermented foods, is intricately shaped by microbial communities, influencing texture, viscosity, and overall structural integrity [16]. The complex matrix changes during fermentation emphasize the profound impact that microbial activities exert on the physical characteristics of the final product. Although individual components such as free amino acids [15], phenolics, volatiles [17], and microstructure [18] have been studied in isolation, a comprehensive understanding of their interdependence is crucial for unraveling the holistic nature of fermented jujube puree development. Variations in microbial strains, environmental conditions, and raw material compositions contribute to the diversity observed in fermented products across cultures and regions, offering a rich scientific exploration and optimization avenue. Despite the longstanding cultural significance of fermented foods, a critical gap exists in understanding the nuanced interrelationships among cellular profile, antioxidant properties, color, free amino acids, phenolics,

volatiles, and microstructure during the lactic–acetic acid fermentation process of jujube fruit, typically its puree.

Consequently, this investigation evaluated the impact of specific strains of indigenous Asian lactic and acetic acid bacteria on the composition of antioxidants, color, free amino acids, polyphenolic profile, volatiles, and microstructure in Junzao jujube puree. Additionally, the study aimed to elucidate the primary phenolic groups and free amino acids responsible for the volatile characteristics and assess the microstructural alterations in the resulting jujube puree subjected to lactic–acetic acid co-fermentation.

## 2. Experimental Procedures

### 2.1. Microbial Isolates, Chemicals and Reagents

Lactic acid bacteria (LAB), including *Lactiplantibacillus plantarum* Lp-28, *Lacticaseibacillus casei* Lc-122, and *Lactobacillus helveticus* Lh-43, were acquired from Synbio Tech Inc. (Kaohsiung City, Taiwan). The acetic acid bacteria (AAB) strain *Acetobacter pasteurianus* (Ap-As.1.41, HuNiang 1.01) was obtained from Yishui Jinrun Biological Technology Co. Ltd. (Yishui, Shandong, China). High-performance liquid chromatography (HPLC) grade L-amino acid standards were sourced from Sigma-Aldrich Chemie GmbH in Schnelldorf, Germany. HPLC phenolic standards were purchased from Shanghai Yuanye Biotechnology Co., Ltd. (Shanghai, China). 2-Octanol ($\geq$99.5%) was procured from Macklin Biochemical Technology Co., Ltd. (Shanghai, China). 2,2-azino-bis-3-ethylbenzothiazoline-6-sulfonic acid (ABTS) and 2,2-diphenyl-1-picrylhydrazyl (DPPH) reagents were sourced from Sinopharm Chemical Reagent Co. Ltd. (Shanghai, China). All additional analytical-grade chemicals were acquired from Sinopharm Chemical Reagent in Shanghai, China, without necessitating further purification.

### 2.2. Jujube Fruit Sampling and Activation of Bacterial Starter Cultures

Fully matured and dried fruits of *Ziziphus jujuba* Mill cv. Junzao, characterized by an intense reddish-brown coloration, were procured commercially from a fruit shop in Zhenjiang, Jiangsu Province China, in May 2023. Stringent criteria were applied to select sampled fruits, excluding those with broken cell walls, mold infestation, or a darkish color. The collected fruits underwent a thorough cleansing process involving an initial wash in a 0.02% sodium hypochlorite solution, succeeded by immersion in distilled water to eradicate surface microbial contaminants. Subsequently, the treated and cleansed samples were preserved in sterile plastic film at −40 °C for 96 h prior to the initiation of jujube puree preparation.

The methodology outlined by Kwaw et al. [13] was followed, albeit with significant modifications, to activate the bacterial strains used in this study. The four distinct microorganisms, namely the *Lactiplantibacillus plantarum* Lp-28, *Lacticaseibacillus casei* Lc-122, *Lactobacillus helveticus* Lh-43, and *Acetobacter pasteurianus* Ap-As.1.41 HuNiang 1.01 strains, were individually subjected to activation procedures. *Lp. plantarum* Lp-28 and *Lc. casei* Lc-122 were activated in de Man Rogosa Sharpe (MRS) broth at 37 °C for 24 h. Simultaneously, *L. helveticus* Lh-43 was activated through subculturing in MRS broth at 37 °C for a similar duration. *A. pasteurianus* Ap-As.1.41, HuNiang 1.01 was activated by subculturing in reinforced acetic acid-ethanol (RAE) broth at 30 °C for 24 h. After activation, the cultures were centrifuged at 4000 rpm, 25 °C, for 10 min using a Ruijiang RJ-TDL-50A centrifuge (Ruijiang Analytical Instrument Co., Ltd., Wuxi, China). After supernatant removal, the bacterial cells were washed in a sterile 0.1% NaCl solution. Inoculum concentration was determined using an XB-K-250 hemocytometer (Jianling Medical Device Co., Danyang, Jiangsu, China) and adjusted to 8 log CFU/mL. The resulting suspensions were employed as starter cultures for the fermentation process of Junzao jujube puree.

### 2.3. Jujube Puree Preparation and Fermentation Procedure

Jujube puree was formulated following the protocol described by Li et al. [19], with substantial modifications. Sterile frozen jujube samples were thawed to ambient tempera-

ture and boiled in distilled water (1:5, *w/v*) for 10 min. Following this, ventrally grooved pits were meticulously removed. Subsequently, the jujubes were combined with distilled water in a 1:2 (*w/v*) ratio and homogenized utilizing a kitchen blender (JYLC91T; Joyoung Co., Ltd., Hangzhou, China). The resultant puree, characterized by a pH of 5.13, exhibited high viscosity at 13 °Brix. Consequently, adjustments were made to the pH (using food-grade $Na_2CO_3$) and °Brix (using distilled water), attaining values of 5.5 and 11, respectively. Preceding the fermentation process, the puree was pasteurized at 70 °C for 30 min.

The pasteurized jujube puree was then inoculated with 1% (*v/v*) of each respective inoculant. The resultant mixture was cooled to ambient temperature and subsequently incubated using a rotary shaking incubator (IS-RDD3, Crystal Technology, and Industries, Suzhou, Jiangsu, China) for 48 h at 37 °C during the anaerobic fermentation phase, facilitating the optimal activity of *Lactobacillus* strains. Furthermore, a subsequent incubation period of 144 h at 30 °C was observed during the aerobic fermentation phase, aligning with conducive conditions for *Acetobacter pasteurianus* Ap-As.1.41 HuNiang 1.01 activity, as Xia et al. [8] suggested.

The distinct combinations of starter cultures employed in the formulation of lactic–acetified jujube puree (JP) were as follows:

A.  *Lacticaseibacillus casei* Lc 122-*A. pasteurianus* Ap-As.1.41 HuNiang 1.01 (JLcAp),
B.  *Lactobacillus helveticus* Lh 43-*A. pasteurianus* Ap-As.1.41 HuNiang 1.01 (JLhAp), and
C.  *Lactiplantibacillus plantarum* Lp 28-*A. pasteurianus* Ap-As.1.41 HuNiang 1.01 (JLpAp).

JCON is comprised of sterile purees subjected to pasteurization without inoculation. Each fermentation was conducted separately in 1000 mL Erlenmeyer flasks.

### 2.4. Functionality of JP

### 2.4.1. Microbial Profile of JP

The microbial assay was performed in accordance with the plate count method delineated by Boasiako et al. [20]. In summary, 1 mL of JP was aseptically pipetted into 9 mL of 0.9% saline water, followed by vortexing for 1 min and subsequent serial dilution. Subsequently, 1 mL of the appropriately diluted samples was plated in triplicate on deMan Rogosa Sharpe (MRS) and reinforced acetic acid–ethanol (RAE) agars. The plated samples were incubated at 37 °C and 30 °C, respectively, for 36 h. Microbial enumeration was conducted by counting plates containing 30–300 colonies, and the results were expressed as the logarithm of the average number of total colony-forming units per mL (log CFU/mL).

### 2.4.2. Antioxidant Properties

ABTS Radical Scavenging Activity (ABTS-RSA)

The ABTS radical scavenging activity (ABTS-RSA) was assessed following Boasiako et al. [20] protocol. Briefly, a mixture comprising ABTS (7 mM) and potassium persulfate (4.95 mM) was prepared in a 1:1 (*v/v*) ratio and kept in darkness at 25 °C for 16 h. This solution was then diluted with methanol to achieve an absorbance of 0.822 at 734 nm. Subsequently, 0.06 mL of the JP sample (diluted 1:15, *v/v*) was added to 2.1 mL of the prepared mixture and incubated in darkness for 10 min. The absorbance was measured at 734 nm using a UV spectrophotometer (UV-1600). Ascorbic acid (20–100 µg/mL) served as the reference standard, and the results were expressed in terms of ascorbic acid equivalent (AAE) antioxidant capacity, presented as mg AAE/100 g JP.

DPPH Radical Scavenging Activity (DPPH-RSA)

A slightly modified version of the protocol outlined by Boateng et al. [21] was employed for DPPH-radical scavenging activity (DPPH-RSA). Briefly, a 0.1 mM DPPH solution was prepared using methanol. The DPPH solution was then diluted with 100% methanol until an absorbance value of 0.876 at 520 nm was attained. Subsequently, 2.1 mL of the DPPH solution was combined with 0.06 mL of the fermented sample (diluted 1:15). After thorough vortexing, the mixture was incubated for 30 min at 25 °C in darkness. The absorbance was measured using a UV spectrophotometer (UV-1600) at 520 nm. Ascorbic acid (20–100 µg/mL)

served as the reference standard, and the results were expressed in terms of ascorbic acid equivalent (AAE) antioxidant capacity, presented as mg AAE/100 g JP.

### 2.4.3. Color Assessment

Color attributes, including lightness–darkness (*L\**), redness–greenness (*a\**), and yellowness–blueness (*b\**), were measured using a HunterLab ColorQuest XE Spectrophotometer (Hunter Associates Laboratory, Virginia, USA). The overall difference in color (ΔE) was determined using Equation (1), as described by Boasiako et al. [20]:

$$\Delta\text{E} = \sqrt{\left(L_o^* - L^*\right)^2 + \left(a_o^* - a^*\right)^2 + \left(b_o^* - b^*\right)^2} \tag{1}$$

where $L^*$, $a^*$, $b^*$ represent the color attributes of JP and $L_o^*$; $a_o^*$, $b_o^*$ represent the control (unfermented sample).

### 2.4.4. Free Amino Acid Profiling
Sample Preparation

An amount of 2 g of the sample (JCON and JP) enclosed in a glass vessel was subjected to maceration with 50 mL of 1% sulfosalicylic acid, employing an ultrasonic mixer operating at 100 W for 20 min. Subsequently, the extract was filtrated through a 0.45 μm Whatman filter paper, and the filtered solution underwent additional filtration through a 0.22 μm syringe membrane for chromatographic analysis.

Chromatographic Analysis

Free amino acids were quantified using a Hitachi model L-8900 amino acid analyzer (Hitachi Co. Ltd., Tokyo, Japan) equipped with a column filled with custom Hitachi ion-exchange resin 2622 (dimensions: 4.6 mm × 60 mm, particle size: 5 μm). Mobile phase A (Ninhydrin) exhibited a flow rate of 0.30 mL/min, while mobile phase B (lithium citrate buffer) was set at 0.35 mL/min. The injection volume was maintained at 20 μL, and column temperatures varied between 30 °C and 70 °C, with the reaction coil reaching a temperature of 135 °C. The amino acid and the jujube puree contents were baseline separated, as shown in Figure S2. The amino acid content was determined using the Equation (2), as specified by Song et al. [15]:

$$X_i = \frac{c \times f \times V \times M}{m \times 10^9} \times 100 \tag{2}$$

$X_i$: content of amino acid "*i*" in samples (g/100 g); *c*: concentration of amino acid "*i*" in solution (nmol/mL); *f*: dilution coefficient; *V*: volume of the fixed specimen (mL); *M*: molar mass of amino acid "*i*" (g/mol); and *m*: mass of the sample (g).

### 2.4.5. Phenolic Profile
Sample Preparation

The protocol described by Dou et al. [22] was followed with minor changes. Briefly, 1 g of each JP and JCON was extracted using 20 mL of ethanol (80%, *v/v*), followed by ultrasonication in a water bath for 30 min. Subsequently, the extract was filtrated through a 0.45 μm Whatman filter paper, and the filtered solution was additionally filtered through a 0.22 μm syringe membrane into dark HPLC glass vials. Additionally, standard phenolic stock solutions (0.1, 31.25, 62.5, 125, 250, and 500 μg/mL) were prepared via dilution in methanol (HPLC grade).

HPLC-UV Detection

The phenolic fraction was determined using a Shimadzu LC 20A system (Shimadzu Incorporated, Tokyo, Japan) according to a method previously reported by Dou et al. [22] with slight modifications. The analytes of interest were eluted on an Agilent ZORBAX-SB C-18 column (4.6 mm × 250 mm, 5 μm particle size, Agilent Technology, Santa Clara, CA, USA).

The mobile phases comprised 0.1% acetic acid in HPLC grade water (mobile phase A) and 100% acetonitrile, HPLC grade (mobile phase B). The linear gradient protocol was set as follows: 0–10 min, 5–10% mobile phase B; 10–15 min, 10–20% mobile phase B; 15–25 min, 20–38% mobile phase B; 25–30 min, 38–40% mobile phase B; 30–31 min, 40–100% mobile phase B; 31–35 min, 100% mobile phase B; 35–36 min, 100–5% mobile phase B; 36–50 min, 5% mobile phase B. The flow rate was kept constant at 0.8 mL/min, and the chromatograms were recorded at 260, 360, and 520 nm for phenolic acids, flavonoids, and anthocyanins, respectively. The column temperature was 30 °C, and the injection volume was 10 µL.

Qualitatively, phenolic compounds were determined by comparing their retention times and absorption spectra with those of the pure standards and quantitatively by peak areas using the pure standards calibration curves. The phenolic compound was expressed as milligrams per 100 g fresh weight of JP (mg/100 g FW).

### 2.5. Volatile Analysis

2.5.1. Elucidation of Volatile Compounds Using HS-SPME-GC-MS

2.5.2. Sample Extraction

The extraction process was performed using the headspace solid-phase microextraction (HS-SPME) technique, adhering to the procedural guidelines outlined by Kwaw et al. [23]. For HS-SPME, a 5 mL sample was introduced into a 15 mL glass vial containing 1.5 g of NaCl and supplemented with an internal standard (ISD) (10 µL of 800 µg/L 2-octanol). The system was equilibrated at 40 °C for 20 min, following which the silicone-septum-sealed vial was hermetically sealed. Subsequently, a divinylbenzene/carboxy/ polydimethylsiloxane (DVB/CAR/PDMS) fiber (50/30 µm) (Fisher Scientific Co. LLC, Pittsburgh, Pennsylvania, USA) was exposed into the headspace of the glass vial for 30 min, and the solution was continuously stirred at a frequency of 2.5 Hz throughout the entire exposure duration.

2.5.3. HS-SPME-GC-MS Column Analysis

Following the extraction process, the fiber was immediately introduced into the injection port of the GC-MS system, comprising an Agilent 6890N-5973B gas chromatograph (Agilent Technologies, Santa Clara, CA, USA) coupled with a mass spectrometer detector. The GC column was Agilent J&W DBWAX (60 m × 0.25 mm × 0.25 µm film thickness, Agilent Technologies). Chromatographic conditions were set as follows: splitless injection mode with injection and detection temperatures of 250 °C, helium utilized as the carrier gas at a flow rate of 1 mL/min, and a temperature program initiated at 50 °C for 10 min, followed by an increase to 150 °C at 6 °C/min, further elevated to 200 °C at 8 °C/min, and maintained for 7 min. Mass spectrometry (MS) was performed with a 23 °C ion source, a 150 °C quadrupole, and an electron impact ionization tune of 70 eV, scanning a range from 33 to 350 atomic mass units. The GC chromatogram is presented in Figure S3.

A prioritized focus was placed on volatile compounds [12,15], with a criterion of over 85% match [15]. Qualitative identification of volatile compounds involved a comparison of their retention indices (RI) and mass spectra with those stored in the National Institute of Standards and Technology (NIST) 17 library database (version 4.52, Shimadzu, Kyoto, Japan) of the GC-MS data system. For quantification of volatile compounds, chromatographic-grade 2-octanol served as an internal standard, with response and calibration factors assumed to be 1.0 [16,17].

$$VC \left( \frac{\text{ng}}{\text{g}} \right) = \frac{peak\ area\ ratio \times 10\ \text{µL}\ (ISD) \times 0.8 \left( \frac{\text{ng}}{\text{µL}} \right) (ISD)}{equivalent\ mass\ of\ volume\ used\ (JP)} \tag{3}$$

2.5.4. Aroma Profile Using Electronic Nose

The methodology established by Chen et al. [24] was implemented for the analysis of aroma profile using the electronic nose (E-nose) PEN-3.5 (Airsense Analytics Inc., Schwerin, Germany), featuring ten metal oxide semiconductor sensors operated at room temperature.

The E-nose, designed for analyzing the headspace of both liquids and solids, as described by Wahia et al. [25], involved the placement of 5 mL of samples within a 25 mL headspace glass vial with a Teflon/silicon septum in the screw lid. A Luer lock needle attached to a 3 mm Teflon tube was utilized to puncture the vial seal, absorbing the air inside the vial and 3 mm above the surface of the samples.

Analytical conditions encompassed a 180 s sensor cleaning period, 10 s automatic zero adjustments, and internal and inlet flow rates set at 600 mL/min each. Detection time was designated as 60 s. The sensor signals, including W1C (aromatics), W5S (broad-range), W3C (aromatic), W6S (hydrogen), W5C (aromatic-aliphatic), W1S (broad methane), W1W (sulfur-organic), W2S (broad-alcohol), W2W (sulfur-organic), and W3S (methane-aliphatic), recorded during the E-nose analysis were systematically computerized and analyzed. The data were captured every second throughout the entire measurement period of the E-nose, based on the sensor matrix data methodology outlined by Bonah et al. [26].

### 2.6. Microstructural Analysis with Scanning Electron Microscopy (SEM)

Microstructural analysis of unfermented and bicultured purees was performed using a JSM-7001F Scanning Electron Microscope (Jeol Ltd., Japan). The investigation was done in low vacuum mode, using a gaseous secondary electron detector (GSED) and maintaining the conditions: an accelerating voltage of 10 kV, 3.5 spot size, 30 μm objective aperture, and 8 mm working distance. Preceding the imagery, the freeze-dried samples were meticulously affixed to a double sticky film on an aluminum metal stub, covering an average area of 1 cm$^2$ [27]. The acquisition of microscopic imagery was facilitated at 1000× magnification levels using an integrated light-color optical navigation camera embedded within the SEM system.

### 2.7. Statistical Analysis

The experiments were replicated three times, and the results were expressed as the mean ± standard deviation. Statistical comparisons were performed using one-way analysis of variance (ANOVA) to determine mean distinctions. Tukey's test assessed statistical significance when the *p*-value was below 0.05. Pearson's correlations were executed to identify and describe relationships among the chosen parameters. The statistical analyses were conducted utilizing Minitab version 18 (Minitab, LLC, Chicago, IL, USA), and OriginPro2021 software (OriginLab®, Northampton, MA, USA). OriginPro2021 software (OriginLab®, Northampton, MA, USA) was specifically utilized for principal component analysis, radar plot generation, and other graphical representations.

### 3. Results and Discussion
#### 3.1. Viable Counts of Lactic and Acetic Acid Bacteria

Evaluating the viability of lactic acid bacteria (LAB) and acetic acid bacteria (AAB) in bicultured jujube puree is pivotal in determining the efficacy of the matrix and its support for lactic–acetic co-fermentation. The data in Table 1 detail the proliferation of LAB and AAB strains in jujube purees both before and after fermentation. Before fermentation, the cell concentration of all selected commercial LAB and AAB strains was approximately 8 log CFU/mL. Post-fermentation, the four strains recorded a substantial increase in viable counts. Of particular note was the significant proliferation of the *Lactobacillus helveticus* (Lh 43) strain in puree JLhAp, reaching a remarkable cell count of 12.20 log CFU/mL. Conversely, *Lacticaseibacillus casei* (Lc 122) and *Lactiplantibacillus plantarum* (Lp 28) showed no statistically significant differences (*p* < 0.05). Furthermore, the AAB cell count in JLpAp was the highest, with no significant differences (*p* < 0.05) compared to matrices JLcAp and JLhAp, indicating a consistent cell concentration of *Acetobacter pasteurianus* (Ap-As.1.41) throughout the formulation. This consistency emphasizes the excellent formulation of the jujube puree, corroborated by similar findings in Muzao juice with *L. acidophilus* at 11.92 log CFU/mL and in Hetian juice with *L. helveticus* and *Lp. plantarum* at 11.86 log CFU/mL. In both cases, viable counts of four LAB strains exceeded 11 log CFU/mL

after fermentation [19]. The control group, JCON, exhibited cell counts of less than 1, suggesting effective elimination of unwanted background during pasteurization that could have interfered with the proliferation of targeted strains (*Lc. casei* Lc 122, *L. helveticus* Lh 43, *Lp. plantarum* Lp 28, and *A. pasteurianus* Ap-As.1.41) in the prepared jujube matrix. Fortunately, post-fermentation, viable counts of lactic and acetic acid bacteria in all jujube purees remained above 6.0–7.0 log CFU/mL, a level known to function as probiotics and promote health, in compliance with the recommended minimum counts of 6.0 log CFU/mL in fermented products [19]. These findings accentuate the potential of jujube puree as an excellent carrier for lactic–acetic co-fermentation studies, promising probiotic functionality and potential health benefits for consumers.

**Table 1.** Viable cell counts, antioxidant properties, and overall color difference of bicultured Junzao jujube puree, JP.

| Parameters | JP | | | |
|---|---|---|---|---|
| | **JLcAp** | **JLhAp** | **JLpAp** | **JCON** |
| **Cell count (log$_{10}$ CFU/mL):** | | | | |
| Before fermentation: LAB | 8.41 ± 0.19 [a,b] | 8.83 ± 0.14 [a] | 8.85 ± 0.23 [a] | <1 [c] |
| AAB: | 8.87 ± 0.31 [a] | 8.87 ± 0.31 [a] | 8.87 ± 0.31 [a] | <1 [b] |
| After fermentation: LAB | 11.10 ± 0.11 [b] | 12.20 ± 0.17 [a] | 11.39 ± 0.42 [b] | <1 [c] |
| AAB | 11.81 ± 0.07 [b] | 12.07 ± 0.15 [a,b] | 12.26 ± 0.19 [a] | <1 [c] |
| **Antioxidant properties:** | | | | |
| ABTS-RSA (mg AAE/100 g FW) | 33.41 ± 0.13 [c] | 34.52 ± 0.07 [b] | 34.61 ± 0.04 [b] | 36.70 ± 0.07 [a] |
| DPPH-RSA (mg AAE/100 g FW) | 34.70 ± 0.04 [d] | 35.68 ± 0.04 [c] | 35.96 ± 0.04 [b] | 38.15 ± 0.04 [a] |
| **Colorimetric:** | | | | |
| ΔE | 4.99 ± 2.41 [b] | 9.66 ± 1.69 [a] | 3.07 ± 0.62 [b,c] | - |

Data expressed as mean ± standard deviation. Means in the same row with different superscript letters are significantly different (*p* < 0.05). **Note:** CFU—colony forming unit; LAB—lactic acid bacteria; AAB—acetic acid bacteria; ABTS-RSA—2,2-azino-bis-3—ethylbenzothiazoline-6-sulfonic acid radical scavenging activities in milligram ascorbic acid equivalent per 100 g fresh weight; DPPH-RSA—2,2-diphenyl-1-picrylhydrazyl radical scavenging activities in milligram ascorbic acid equivalent per 100 g fresh weight; ΔE—the overall difference in color. JLcAp—*Lacticaseibacillus casei* Lc 122—*Acetobacter pasteurianus* Ap-As.1.41 HuNiang 1.01 puree; JLhAp—*Lactobacillus helveticus* Lh 43—*A. pasteurianus* Ap-As.1.41 HuNiang 1.01 puree; JLpAp—*Lactiplantibacillus plantarum*—*A. pasteurianus* Ap-As.1.41 HuNiang 1.01 puree; (JCON)—puree with no bacteria inoculants (unfermented).

### 3.2. Antioxidant Properties

To thoroughly assess the overall antioxidant capacity of bicultured puree, a range of assays were employed, recognizing the limitations of each in capturing specific antioxidants [15,28]. The results, detailed in Table 1, showed the outcomes of these various assays. Remarkably, the control group (unfermented) exhibited the highest antioxidant activities, particularly in ABTS-RSA and DPPH-RSA, compared to various bicultured puree treatments. This aligns with expectations, considering ascorbic acid's status as a vitamin, potentially utilized by bacterial strains due to their energy requirements. The lower antioxidant activities could also be attributed to increased compound migration, resulting in bioactive compounds' leaching and a subsequent reduction in antioxidant activity [29]. Among the pretreated samples, both JLhAp and JLpAp recorded the highest ABTS antioxidant activity, with no significant differences (*p* < 0.05) compared to JLcAp. ABTS-RSA and DPPH-RSA showed a remarkably high significant positive correlation (*r* = 0.996, Table S1). It is important to note that the employed bacterial strains in the study possess unique characteristics and distinct systems for microbial activities, including the elicitation of hydrolytic enzymes for macromolecular disintegration [13]. A Pearson correlation analysis was conducted to gain deeper insights into antioxidant capacity and individual phenolic relationships (Table S1). Strong positive correlations were observed between phenolic compounds (chlorogenic acid, rutin, *p*-coumaric acid) and antioxidant capacities (ABTS-RSA and DPPH-RSA), indicating that an increase in chlorogenic acid, rutin, and *p*-coumaric acid corresponds to an increase in antioxidant activity. This find-

ing aligns with previous research on ginger slices and coffee leaves [25,26]. Conversely, sinapic acid, ferulic acid, and gallic acid exhibited very negative relations, implying that increases in sinapic acid, ferulic acid, and gallic acid were associated with a significant reduction in antioxidant properties. This emphasizes the significant contribution of specific compounds to the radical scavenging ability in the bicultured jujube purees, highlighting that an increase in chlorogenic acid, rutin, and *p*-coumaric acid corresponds to heightened antioxidant properties (DPPH-RSA and ABTS-RSA).

### 3.3. Effect of Lactic–Acetic Acid Co-Fermentation on Color

LAB fermentation's impact on the color attributes of jujube juices was assessed by evaluating total color difference (ΔE), as detailed in Table 1. Notably, significant differences in ΔE were observed among bicultured purees, particularly fermented jujube puree. JLhAp exhibited a larger ΔE compared to fermented JLcAp and JLpAp purees, suggesting that *Lactobacillus helveticus* Lh-43, with minimal inhibitory influence from *Acetobacter pasteurianus* Ap-As.1.41, effectively modified the darker color of the raw jujube puree [30] in its bicultural form. Given that mulberry fruit juice primarily stems from anthocyanins [13], it was also imperative to identify the anthocyanins responsible for the coloration of bicultured Junzao jujube purees. The higher overall color difference observed in the fermented samples (JLcAp, JLhAp, and JLpAp) compared to JCON (Table 1) could be attributed to the increased concentration of monomeric anthocyanins [13] in JP. This was substantiated by a significant ($p < 0.05$) negative correlation (Table S1) between ΔE and cyanidin 3-O-rutinoside ($R^2 = -0.768$), implying that lactic acid fermentation led to a modified yellow transparent jujube puree. These findings align with earlier reports on the impact of fermentation on color attributes [30]. Moreover, the unique metabolic activities of the bacterial strains resulted in a slightly noticeable differential chromatic range of the ΔE values for JP and was observed to fall within a slightly noticeable range of $10.0 < \Delta E < 3.0$ [31], making the color variation more visible [13]. Additionally, Pearson correlation analysis conducted on ΔE revealed positive correlations between the overall color difference and individual phenolic compounds: sinapic acid, ferulic acid, gallic acid, neochlorogenic acid, caffeic acid, protocatechuic acid, catechin, rutin, syringic acid, and 2,3,4-trihydroxybenzoic acid. Conversely, a strong negative correlation existed between ΔE and *p*-coumaric acid. Consequently, enhancing jujube color during lactic–acetic acid fermentation holds promise for improving biotechnological functionality and organoleptic quality [13,32].

### 3.4. Assessment of Free Amino Acid Content

Table 2 illustrates the free amino acid (FAA) composition of bicultured Junzao jujube purees. In comparison to the control (184.03 ± 1.16 mg/100 g), the total free amino acids exhibited variability, with the highest content observed in JLhAp (157.17 ± 1.12 mg/100 g), intermediate levels in JLpAp (153.11 ± 1.80 mg/100 g), and the lowest in JLcAp (139.62 ± 1.15 mg/100 g). Amino acids play a pivotal role in protein synthesis and contribute to the flavor profile of food [33]. While previous studies [8,23] reported high protein amounts in jujube fruits, the lower quantities in the bicultured Junzao purees may be attributed to molecular changes—such as protein cross-linking [34,35] facilitated by various enzymes, like tyrosinases, during drying [36]—and partially utilized by the LABs and AAB during biotransformation owing to their fastidious nature. Among the detected FAAs, L-aspartic acid and proline emerged as the most abundant, averaging 16.35% and 56.28%, respectively, of the total free amino acids in the bicultured and control purees. The initially high content of L-proline in the unfermented samples (111.70 ± 0.10 mg/100 g), constituting around 60.7% of the total FAAs in the control, diminished after fermentation. This reduction may be attributed to significant molecular modifications initiated by LABs and AAB through their hydrolytic enzymes, including (poly)-phenol oxidases [17], esterases [13], and β-glucosidases [37]. These enzymes depolymerized macromolecular phytochemicals, such as isoflavone β-glycosides, into simpler forms (aglycones), leading to structural degradation in L-proline. Distinctively, JLpAp demonstrated the synthesis of

L-methionine (0.85 ± 0.09 mg/100 g), a free amino acid not found in the control (JCON), JLcAp, or JLhAp. This suggests that *Lactiplantibacillus plantarum* Lp 28 possesses a methionine synthetase system, endowing JLpAp with eight essential free amino acids (histidine, isoleucine, leucine, lysine, methionine, phenylalanine, threonine, and valine), while the control, JLcAp, and JLhAp remained methionine limiting. Furthermore, glutamic acid was found at markedly high concentrations in the bicultured purees (~5.49%) compared to the reference sample (3.77%), possibly contributing significantly to volatile hydroxy acids, including acetic acid, hexanoic acid, and octanoic acid (Tables 4 and S2), largely under the influence of aminotransferase [9]. In conclusion, the free amino acids in bicultured Junzao jujube purees were significantly impacted by the intricate reciprocal action of lactic and acetic bacteria in one excellent food matrix—jujube puree.

**Table 2.** Free amino acid content in bicultured Junzao jujube purees.

| Amino Acids | JP (mg/100 g) | | | |
|---|---|---|---|---|
| | JLcAp | JLhAp | JLpAp | JCON |
| L-Aspartic acid | 23.35 ± 0.05 [d] | 26.45 ± 0.11 [c] | 27.46 ± 0.11 [a] | 27.10 ± 0.10 [b] |
| L-Threonine | 2.00 ± 0.05 [b] | 2.41 ± 0.05 [a] | 1.93 ± 0.05 [b] | 2.30 ± 0.10 [a] |
| L-Serine | 2.85 ± 0.11 [b] | 3.47 ± 0.00 [a] | 3.03 ± 0.09 [b] | 3.30 ± 0.10 [a] |
| L-Glutamic acid | 7.57 ± 0.05 [c] | 8.76 ± 0.11 [a] | 8.37 ± 0.14 [b] | 6.93 ± 0.06 [d] |
| L-Glycine | 2.03 ± 0.05 [c] | 2.57 ± 0.11 [b] | 2.09 ± 0.09 [c] | 3.20 ± 0.10 [a] |
| L-Alanine | 3.82 ± 0.05 [c] | 4.23 ± 0.09 [a,b] | 4.01 ± 0.14 [b,c] | 4.40 ± 0.10 [a] |
| L-Valine | 2.60 ± 0.11 [b] | 4.82 ± 0.11 [a] | 1.99 ± 0.09 [c] | 2.83 ± 0.06 [b] |
| L-Methionine | nd | nd | 0.85 ± 0.09 [a] | nd |
| L-Isoleucine | 1.41 ± 0.19 [c] | 1.84 ± 0.07 [b] | 1.23 ± 0.09 [c] | 3.70 ± 0.10 [a] |
| L-Leucine | 2.32 ± 0.05 [b] | 2.82 ± 0.00 [a] | 2.15 ± 0.05 [c] | 2.93 ± 0.06 [a] |
| L-Tyrosine | 1.00 ± 0.05 [a,b] | 1.15 ± 0.10 [a] | 0.85 ± 0.01 [b] | 0.83 ± 0.06 [b] |
| L-Phenylalanine | 2.57 ± 0.11 [a] | 2.66 ± 0.05 [a] | 0.76 ± 0.00 [c] | 1.17 ± 0.06 [b] |
| L-Histidine | 5.51 ± 0.05 [c] | 6.23 ± 0.11 [a] | 6.00 ± 0.14 [a,b] | 5.93 ± 0.06 [b] |
| L-Lysine | 2.75 ± 0.05 [c] | 5.10 ± 0.05 [a] | 2.91 ± 0.05 [b] | 5.17 ± 0.06 [a] |
| L-Arginine | 2.19 ± 0.05 [c] | 2.82 ± 0.09 [a] | 2.59 ± 0.05 [b] | 2.53 ± 0.06 [b] |
| L-Proline | 77.65 ± 0.09 [d] | 81.85 ± 0.05 [c] | 86.89 ± 0.55 [b] | 111.70 ± 0.10 [a] |
| **Total free amino acids** | **139.62 ± 1.15 [d]** | **157.17 ± 1.12 [b]** | **153.11 ± 1.80 [c]** | **184.03 ± 1.16 [a]** |

Data expressed as mean ± standard deviation. Means in the same row with different superscript letters are significantly different (*p* < 0.05). **Note:** n.d—not detected. JLcAp—*Lacticaseibacillus casei* Lc 122—*Acetobacter pasteurianus* Ap-As.1.41 HuNiang 1.01 puree; JLhAp—*Lactobacillus helveticus* Lh 43—*A. pasteurianus* Ap-As.1.41 HuNiang 1.01 puree; JLpAp—*Lactiplantibacillus plantarum*—*A. pasteurianus* Ap-As.1.41 HuNiang 1.01 puree; (JCON)—puree with no bacteria inoculants (unfermented).

### 3.5. Effect of Lactic–Acetic Acid Co-Fermentation on Phenolics of Bicultured Jujube Purees

In the unfermented samples, the primary phenolic compounds—namely chlorogenic acid, sinapic acid, quercetin, catechin, and peonidin-3,5-diglucoside—were quantified at approximately 1.29 mg/100 g, 0.85 mg/100 g, 1.29 mg/100 g, 1.60 mg/100 g, and 25.19 mg/100 g, respectively. Following the fermentation treatment, a remarkable and statistically significant (*p* < 0.05) augmentation in the concentrations of key phenolic compounds was observed in the bicultured purees, specifically sinapic acid, ferulic acid, and rutin, reaching 243%, 210%, and 239% of their respective initial values on average. Notably, phenolic compounds such as ferulic acid are reported to undergo reduction, resulting in the formation of dihydroferulic acid. Despite the presence of decarboxylases and reductases in LAB strains, exemplified by *Lactiplantibacillus plantarum* GK3 and *Lactobacillus acidophilus* 85 [17,18,38], the observed increase in ferulic acids in the bicultured purees contradicts this phenomenon. This apparent contradiction supports the hypothesis that lactic–acetic acid co-fermentation can enhance specific phenolic compounds, such as ferulic acid. Lactic acid bacteria (LAB) are known to enzymatically catalyze the cleavage and subsequent acidification of glycosides and esters conjugated with phenolic compounds. Also, acetic acid

bacteria (AAB) produce acetic acid that can penetrate cell membranes and cause alterations in normal fundamental physiological functions [15].

Significantly, the natural prevalence of conjugates of *p*-coumaric acid in the unfermented sample surpassed that of their fermented forms. Post-fermentation, the reduction in *p*-coumaric acid concentrations could be attributed to the inefficient hydrolysis of corresponding *p*-coumarate esters and *p*-coumarylated anthocyanins inherent in jujube samples [39]. These phenolic acids might have been encapsulated in the undigested dietary fiber content, reducing their extractability [40] and subsequent quantification. Additionally, the subsequent evolution of caffeic acid in JLcAp and JLhAp could be explained by the intricate enzymatic cascade facilitating the conversion of *p*-coumaric acid into caffeic acid, orchestrated by the catalytic prowess of *p*-coumarate 3-hydroxylase [41]. Moreover, the dynamic enzymatic reciprocity includes the transformation of chlorogenic acid into caffeic acid mediated by cinnamoyl esterase, explaining the observed increase in caffeic acid and simultaneous decrease in chlorogenic acid following the lactic–acetic co-fermentation process [17]. The metabolic activities of LAB, encompassing the intricate transformation of sugars, organic acids, and amino acids, profoundly influence the phenolic profile. Substantially, the amino acid phenylalanine undergoes a complex enzymatic transformation into *p*-coumaric acid through the orchestrated action of phenylalanine ammonia-lyase and cinnamate 4-hydroxylase. These intricate biochemical reactions appeared to be operative in the context of *Streptococcus thermophilus* fermentation [19]. As shown in Table 3, the fermentation process resulted in a significant ($p < 0.05$) reduction in the concentrations of chlorogenic acid. The intricate metabolic landscape of LAB is further elucidated by their capacity to metabolize phenolic acids through strain-specific decarboxylase or reductase activities [18]. In this context, chlorogenic acids are envisaged to undergo decarboxylation mediated by phenolic acid decarboxylase, yielding alternative compounds such as 4-vinyl guaiacol and guaiacol. Although Pan et al. [18] reported a decrease in caffeic acid and syringic acid, their report contrasts our findings, as caffeic acid concentrations were enhanced in bicultured purees: JLcAp and JLhAp, whereas syringic acid, undetected in the control sample, evolved after fermentation. The evolution of phenolic compounds, namely gallic acid, protocatechuic acid, and syringic acid, that were undetected in the control sample, forecasts the efficiency of lactic–acetic acid bacteria's intricate microbial synergy in enhancing these bioactives in the fermented product. Morin concentration was enhanced by 151% of the initial value in bicultured puree containing *Lacticaseibacillus casei*, Lc 122, and *Acetobacter pasteurianus* Ap-As.1.41, whereas quercetin increased by 121%. These two flavonol molecules were noticeably absent in the bicultures JLhAp and JLpAp, highlighting the unique ability of *Lc. casei* Lc 122 to exclusively contribute to the expression of morin and quercetin in lactic–acetic co-fermentation. Consequently, this predisposed puree JLcAp to be the richest in total flavonoids, followed by JLhAp. Regarding anthocyanins, high concentrations were observed, with the predominant classes being peonidin-3, 5-diglucoside, and peonidin-3-O-glucoside. The bicultured puree JLcAp recorded the highest peonidin-3,5-diglucoside content at 34.13 mg/100 g, whereas peonidin-3-O-glucoside measured 15.12 mg/100 g. Although cyanidin-3-O-rutinoside was observed to be the lowest among the anthocyanins in the bicultured purees, JLcAp exhibited the highest characteristic content compared to other fermented purees and the control sample. This wide variability in anthocyanin content could be attributed to the unique enzymatic capabilities of LAB and AAB in the fermentation study, eliciting specific hydrolytic enzymes that affect the stability of these anthocyanins in the jujube puree matrix. These compounds are notably influenced by factors such as pH and methylation or acylation at the hydroxyl groups on the A and B rings of the anthocyanin backbone [42]. Collectively, these multifaceted findings emphasize the dynamic and strain-specific enzymatic transformations orchestrating the bioconversion of phenolic compounds during lactic–acetic acid co-fermentation of fruits.

**Table 3.** Phenolic profile of bicultured Junzao jujube purees.

| Phenolic Compounds | MF | RT (min) | Concentration (mg/100 g) | | | |
|---|---|---|---|---|---|---|
| | | | JLcAp | JLhAp | JLpAp | JCON |
| *Phenolic acids* | | | | | | |
| Chlorogenic acid | $C_{16}H_{18}O_9$ | 11.72 | 0.427 ± 0.000 [c] | 0.973 ± 0.000 [b] | 0.385 ± 0.000 [d] | 1.293 ± 0.000 [a] |
| Sinapic acid | $C_{11}H_{12}O_5$ | 19.61 | 2.202 ± 0.000 [a] | 2.011 ± 0.000 [b] | 1.938 ± 0.000 [c] | 0.845 ± 0.000 [d] |
| Ferulic acid | $C_{10}H_{10}O_4$ | 19.66 | 0.731 ± 0.000 [a] | 0.702 ± 0.000 [b] | 0.679 ± 0.000 [c] | 0.335 ± 0.000 [d] |
| Gallic acid | $C_7H_6O_5$ | 6.45 | 0.615 ± 0.000 [c] | 0.633 ± 0.000 [b] | 0.639 ± 0.000 [a] | n.d |
| Neochlorogenic acid | $C_{16}H_{18}O_9$ | 9.63 | 0.357 ± 0.000 [d] | 2.646 ± 0.000 [a] | 1.306 ± 0.000 [b] | 0.747 ± 0.000 [c] |
| Caffeic acid | $C_9H_8O_4$ | 14.47 | 0.513 ± 0.000 [b] | 0.898 ± 0.000 [a] | n.d | 0.235 ± 0.000 [c] |
| Protocatechuic acid | $C_7H_6O_4$ | 9.85 | n.d | 0.091 ± 0.000 [a] | 0.034 ± 0.000 [b] | n.d |
| Syringic acid | $C_9H_{10}O_5$ | 15.19 | 0.023 ± 0.000 [b] | 0.268 ± 0.000 [a] | n.d | n.d |
| 2,3,4-trihydroxybenzoic acid | $C_7H_6O_5$ | 9.10 | 0.147 ± 0.000 [c] | 0.231 ± 0.000 [a] | 0.132 ± 0.000 [d] | 0.168 ± 0.000 [b] |
| *P*-coumaric acid | $C_9H_8O_3$ | 18.56 | n.d | n.d | n.d | 0.057 ± 0.000 [a] |
| Total phenolic acids | | | 5.016 ± 0.001 [b] | 8.453 ± 0.000 [a] | 5.115 ± 0.000 [b] | 3.681 ± 0.000 [c] |
| *Flavonols* | | | | | | |
| Morin | $C_{15}H_{10}O_7$ | 23.51 | 1.088 ± 0.000 [a] | n.d | n.d | 0.721 ± 0.000 [b] |
| Quercetin | $C_{15}H_{10}O_7$ | 23.65 | 1.560 ± 0.000 [a] | n.d | n.d | 1.293 ± 0.000 [b] |
| Catechin | $C_{15}H_{14}O_6$ | 12.57 | 0.779 ± 0.000 [d] | 2.983 ± 0.000 [a] | 1.139 ± 0.000 [c] | 1.600 ± 0.000 [b] |
| Rutin | $C_{27}H_{30}O_{16}$ | 18.37 | 0.987 ± 0.000 [a] | 0.950 ± 0.000 [b] | 0.949 ± 0.000 [c] | 0.402 ± 0.000 [d] |
| Total flavonols | | | 4.416 ± 0.001 [a] | 3.933 ± 0.000 [c] | 2.088 ± 0.000 [d] | 4.016 ± 0.000 [b] |
| *Anthocyanins* | | | | | | |
| Cyanidin-3-O-rutinoside | $C_{27}H_{31}O_{15}$ | 10.64 | 5.146 ± 0.001 [a] | 3.876 ± 0.000 [d] | 4.679 ± 0.000 [c] | 5.102 ± 0.000 [b] |
| Peonidin-3-O-glucoside | $C_{22}H_{23}O_{12}$ | 12.21 | 10.642 ± 0.000 [b] | 6.703 ± 0.000 [d] | 15.119 ± 0.000 [a] | 10.343 ± 0.000 [c] |
| Peonidin-3,5-diglucoside | $C_{28}H_{33}O_{16}$ | 9.94 | 34.133 ± 0.000 [a] | 27.484 ± 0.000 [b] | 19.851 ± 0.000 [d] | 25.193 ± 0.000 [c] |
| Total anthocyanins | | | 49.921 ± 0.001 [a] | 38.063 ± 0.000 [d] | 39.649 ± 0.000 [c] | 40.638 ± 0.000 [b] |
| **Total polyphenol concentration** | | | **59.352 ± 0.002** [a] | **50.450 ± 0.001** [b] | **46.851 ± 0.000** [d] | **48.335 ± 0.000** [c] |

Data expressed as mean ± standard deviation. Means in the same row with different superscript letters are significantly different ($p < 0.05$). **Note:** JLcAp—*Lacticaseibacillus casei* Lc 122—*Acetobacter pasteurianus* Ap-As.1.41 HuNiang 1.01 puree; JLhAp—*Lactobacillus helveticus* Lh 43—*A. pasteurianus* Ap-As.1.41 HuNiang 1.01 puree; JLpAp—*Lactiplantibacillus plantarum*—*A. pasteurianus* Ap-As.1.41 HuNiang 1.01 puree; (JCON)—puree with no bacteria inoculants (unfermented).

### 3.6. Smell Analysis with Electronic Nose (E-Nose)

The electronic nose (E-nose), a pivotal tool for assessing the concentration and presence of odor molecules, plays a significant role in determining product quality and consumer acceptability [20]. The results of the E-nose analysis (Figures 1 and S1, and Table S3) offer insights into the conductivity of sample gas and zero gas to the sensors [43]. Sensor response, denoted by the ratio of conductance (G0/G or G/G0, where G0 and G represent sensor conductance before and after exposure to gas samples, respectively), exhibited diverse trends across different sensors. Sensors WIW (responsive to organic sulfur compounds and terpenes), W5S (broad range), W1S (broad range for methane), and W2S (broad range for alcohols) displayed an initial sharp rise in the first 10 s, followed by a gradual decline to a steady value after 20 s. Conversely, sensors W1C (aromatics), W3C (aromatic), W6S (hydrogen), W5C (aromatic-aliphatic), W2W (sulfur-organic), and W3S (methane-aliphatic) maintained relatively constant values close to 1, indicating minimal responses [15]. The sensorgram output (Figure S1) revealed similarities between the control responses and the bicultures consisting of *Lactobacillus helveticus*, Lh 43, and *Acetobacter pasteurianus*, Ap-As.1.41. Both treatments exhibited a sharp rise in sensors W1W, W5S, W1S, and W2S during the first 10 s, almost maintaining these elevated levels throughout the analysis. Fermented purees, including *Lacticaseibacillus casei* Lc-122 and *A. pasteurianus* Ap-As.1.41, as well as *Lactiplantibacillus plantarum*, Lp-28 and *A. pasteurianus*, Ap-As.1.41, displayed a similar trend with slight modifications. Notably, sensors W1S and W2S showed persistent elevated levels after 20 s, indicating lingering concentrations of these volatiles. The inability of sensors

W1W, W5S, W1S, and W2S to reach nearly zero implies the presence of high concentrations of these volatiles. Variations in bicultured sample responses to sensors may be attributed to cellular-level adaptive mechanisms, inducing hydrolytic enzymes [12,31,44] and releasing bound flavor precursors in raw jujube purees during the lactic–acetic co-fermentation. The robust physiological nature of *A. pasteurianus*, Ap-As.1.41, known to produce acetic acid that can penetrate cell membranes, might have enhanced volatile release [7]. The sensor responses were translated into radar fingerprints (Figure 1a), highlighting that sensors W5S, W1S, W1W, and W2S were most predominant in the bicultured purees compared to the control, JCON. This substantial variation in smell suggests an expansive volatile fraction in the bicultured purees compared to the unfermented sample (Table 4). E-nose parameters of fermented purees (JP) and control underwent principal component analysis, where PC1 (95.88%) and PC2 (2.52%) explained 98.7% of the sample variance (Figure 1b). The control sample exhibited no distinct sensor bioactives but was closely associated with W5C, W3C, and W1C, indicating significant inherent responses from these three sensors, predominantly organic aromatics. All three groups of fermented samples: JLcAp, JLhAp, and JLpAp, appeared to have closely associated parameters (W5S, W6S, W1S, W1W, W2S, W2W, and W3S). The higher number of sensor bioactives associated with the fermented samples could be the basis for the highest volatile concentration profiled (Tables 4 and S2). Although the overall volatile composition of the control was higher ($p < 0.05$) than the fermented samples, it is also reported that certain analytical inefficiencies, such as the method of extraction [40] as well as inherent properties of the food matrix (such as condensed bioactives—proanthocyanidins, tannins) [45] may inhibit the absolute extractability of certain bioactive compounds, such as volatiles or aroma compounds.

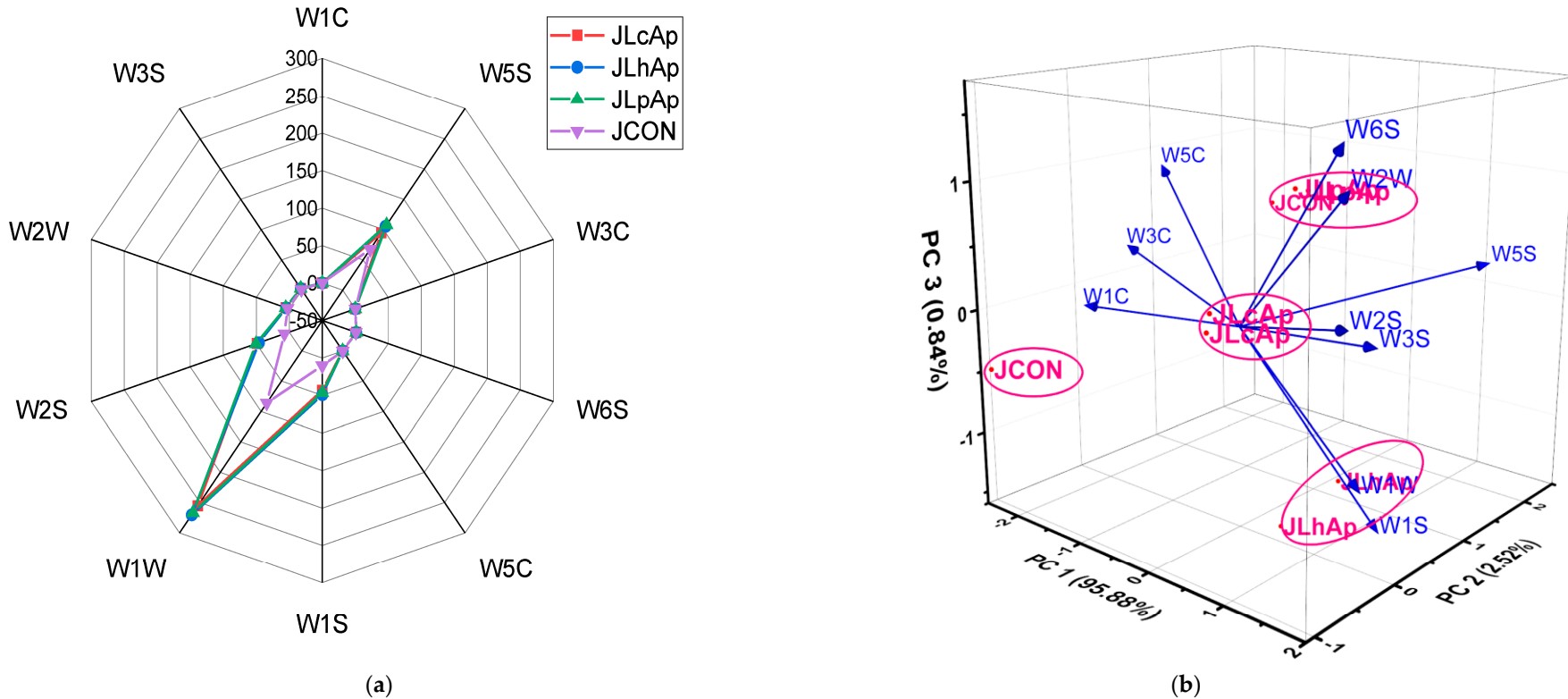

(**a**)          (**b**)

**Figure 1.** Radar plot (**a**) and principal component analysis (**b**) of electronic nose sensor sensitives of bicultured jujube purees. **Note:** W1C—aromatic organic compounds; W5S—broad range sensitivity; W3C—aromatic compounds; W6S—hydrogen gas; W5C—aromatic–aliphatic compounds; W1S—broad methane; W1W—organic sulfur compounds and terpenes; W2S—broad-range alcohols; W2W—aromatic, inorganic sulfur and organic compounds; W3S—methane and aliphatic organic compounds. JLcAp—*Lacticaseibacillus casei* Lc 122—*Acetobacter pasteurianus* Ap-As.1.41 HuNiang 1.01 puree; JLhAp—*Lactobacillus helveticus* Lh 43—*A. pasteurianus* Ap-As.1.41 HuNiang 1.01 puree; JLpAp—*Lactiplantibacillus plantarum*—*A. pasteurianus* Ap-As.1.41 HuNiang 1.01 puree; (JCON)—puree with no bacteria inoculants (unfermented).

**Table 4.** Quantitative amounts of volatile compounds identified in bicultured Junzao jujube purees by HS-SPME-GC/MS method.

| Volatile Groups | SN | Compound Name | CAS Number | Odor Description | Concentration of JP, ng/100 g FW (Mean ± SD) | | | |
|---|---|---|---|---|---|---|---|---|
| | | | | | JCON | JLcAp | JLhAp | JLpAp |
| Alcohols | AL1 | 1-Decanol | 112-30-1 | Floral, fatty | 22.79 ± 0.03 [a] | n.d | n.d | n.d |
| | AL2 | 1-Dodecanol | 112-53-8 | Sweet, fats, coconut | n.d | n.d | 2.22 ± 0.07 [a] | 1.66 ± 0.10 [b] |
| | AL3 | 1-Heptanol | 111-70-6 | Green | n.d | 3.22 ± 0.06 [b] | n.d | 3.39 ± 0.06 [a] |
| | AL4 | 1-Hexanol | 111-27-3 | Resin, flower, green | 3.02 ± 0.08 [a] | 2.34 ± 0.004 [b] | 2.20 ± 0.09 [b,c] | 2.15 ± 0.0 [c] |
| | AL5 | 1-Hexanol, 2-ethyl- | 104-76-7 | Green, rose | 14.51 ± 0.03 [a] | 1.19 ± 0.004 [c] | 1.28 ± 0.03 [b] | 1.10 ± 0.03 [d] |
| | AL6 | 1-Hexanol, 5-methyl- | 627-98-5 | | n.d | n.d | 3.41 ± 0.05 [a] | n.d |
| | AL7 | 1-Nonanol | 143-8-8 | Fatty | 1.24 ± 0.04 [a] | n.d | n.d | n.d |
| | AL8 | 1-Octanol | 111-87-5 | Chemical, metal, burnt | 4.08 ± 0.06 [b] | n.d | n.d | 6.21 ± 0.07 [a] |
| | AL9 | 2,3-Butanediol | 24347-58-8 | Buttery, creamy | 264.05 ± 2.54 [a] | 56.36 ± 0.11 [d] | 65.74 ± 0.03 [b] | 61.71 ± 0.05 [c] |
| | AL10 | 2,4-Di-tert-butylphenol | 96-76-4 | Phenolic | 3.48 ± 0.02 [a] | 1.14 ± 0.04 [d] | 1.83 ± 0.05 [b] | 1.40 ± 0.04 [c] |
| | AL11 | 2-Butanol, 1-methoxy- | 53778-73-7 | | n.d | n.d | 1.36 ± 0.04 [a] | n.d |
| | AL12 | 2-Nonanol | 628-99-9 | Orange, rose, mushroom | 1.24 ± 0.04 [b] | n.d | 2.71 ± 0.04 [a] | n.d |
| | AL13 | Benzyl alcohol | 100-51-6 | Walnut, nutty | 14.95 ± 0.04 [a] | 4.95 ± 0.12 [c] | 5.13 ± 0.05 [c] | 5.42 ± 0.07 [b] |
| | AL14 | Phenol | 108-95-2 | Plastic, rubber | n.d | 6.43 ± 0.14 [b] | n.d | 7.26 ± 0.06 [a] |
| | AL15 | Phenol, 4-ethyl- | 123-7-9 | Phenol | n.d | 4.88 ± 0.13 [a] | n.d | 2.96 ± 0.05 [b] |
| | AL16 | Phenylethyl Alcohol | 60-12-8 | Floral, rosy, honey, spice | 5.09 ± 0.04 [d] | 83.19 ± 0.22 [c] | 90.72 ± 0.04 [a] | 84.93 ± 0.03 [b] |
| | AL17 | Thymol | 89-83-8 | Herb, pleasant | n.d | 13.75 ± 0.05 [a] | n.d | n.d |
| | | **Subtotal** | | | **333.20 ± 2.70 [a]** | **177.43 ± 0.22 [b]** | **176.61 ± 0.06 [b]** | **178.19 ± 0.40 [b]** |
| Acids | ACD1 | 2-Heptenoic acid | 18999-28-5 | Green, fruity | 7.87 ± 0.03 [a] | n.d | n.d | n.d |
| | ACD2 | 2-Octenoic acid | 1470-50-4 | | n.d | 1.11 ± 0.03 [a] | 0.89 ± 0.10 [b] | n.d |
| | ACD3 | 3-Decenoic acid, (E)- | 53678-20-9 | | n.d | 1.00 ± 0.05 [b] | 1.36 ± 0.10 [a] | n.d |
| | ACD4 | 3-Hexenoic acid, (E)- | 1577-18-0 | Must, fat | 2.09 ± 0.03 [a] | n.d | n.d | n.d |
| | ACD5 | 3-Octenoic acid, (E)- | 5163-67-7 | | 3.23 ± 0.06 [a] | n.d | 2.15 ± 0.06 [b] | n.d |
| | ACD6 | Benzoic acid | 65-85-0 | Leather | 6.33 ± 0.01 [b] | n.d | 19.86 ± 0.07 [a] | 6.42 ± 0.07 [b] |
| | ACD7 | Benzoic acid, p-tert-butyl- | 98-73-7 | | 1.48 ± 0.03 [a] | n.d | n.d | n.d |
| | ACD8 | Butanoic acid, 2-methyl- | 116-53-0 | Cheesy, sweaty | 2.04 ± 0.03 [a] | n.d | n.d | n.d |
| | ACD9 | Butanoic acid, 3-methyl- | 503-74-2 | Rancid, cheesy, sweaty | 1.73 ± 0.03 [a] | n.d | n.d | n.d |

**Table 4.** *Cont.*

| Volatile Groups | SN | Compound Name | CAS Number | Odor Description | Concentration of JP, ng/100 g FW (Mean ± SD) | | | |
|---|---|---|---|---|---|---|---|---|
| | | | | | JCON | JLcAp | JLhAp | JLpAp |
| | ACD10 | Dodecanoic acid | 143-7-7 | Rancid, moldy | 15.03 ± 0.01 [a] | 11.54 ± 0.03 [c] | 11.62 ± 0.05 [c] | 11.83 ± 0.04 [b] |
| | ACD11 | Heptanoic acid | 111-14-8 | Rancid, fatty | 4.04 ± 0.03 [c] | 4.52 ± 0.09 [b] | 11.54 ± 0.07 [a] | 3.71 ± 0.03 [d] |
| | ACD12 | Hexanoic acid | 142-62-1 | Cheesy, fatty | 48.76 ± 0.01 [a] | 6.63 ± 0.08 [b] | n.d | n.d |
| | ACD13 | Hydrocinnamic acid | 501-52-0 | Cheesy | 3.96 ± 0.06 [b] | 3.40 ± 0.05 [c] | 3.06 ± 0.06 [d] | 4.25 ± 0.06 [a] |
| | ACD14 | n-Decanoic acid | 334-48-5 | Fatty, citrus | 43.79 ± 0.04 [a] | 7.21 ± 0.13 [c] | 9.16 ± 0.09 [b] | 5.55 ± 0.06 [d] |
| | ACD15 | Nonanoic acid | 112-5-0 | Waxy, cheese-like | 14.42 ± 0.02 [a] | 10.45 ± 0.08 [b] | n.d | 3.34 ± 0.06 [c] |
| | ACD16 | Octanoic acid | 124-7-2 | Fatty, rancid | 24.43 ± 0.09 [a] | 10.98 ± 0.15 [c] | 22.27 ± 0.07 [b] | 9.72 ± 0.03 [d] |
| | ACD17 | Pentanoic acid | 109-52-4 | Sweet | 4.77 ± 0.03 [a] | 2.17 ± 0.03 [b] | n.d | n.d |
| | ACD18 | Pentanoic acid, 3-methyl- | 105-43-1 | Cheesy, fruity | 4.51 ± 0.01 [a] | n.d | n.d | n.d |
| | ACD19 | Tetradecanoic acid | 544-63-8 | Waxy, fatty, coconut | 2.01 ± 0.06 [a] | n.d | n.d | n.d |
| | | **Subtotal** | | | **190.49 ± 0.51 [a]** | **59.01 ± 0.39 [c]** | **81.90 ± 0.18 [b]** | **44.82 ± 0.19 [d]** |
| Ketones | KTN1 | 2-Pentanone, 5-methoxy- | 17429-4-8 | | 8.66 ± 0.10 [a] | n.d | n.d | n.d |
| | KTN2 | 2-Hydroxy-3-hexanone | 54073-43-7 | | 7.32 ± 0.04 [a] | n.d | n.d | n.d |
| | KTN3 | 2-Nonanone | 821-55-6 | Vegetable, moldy | 12.66 ± 0.06 [a] | n.d | n.d | n.d |
| | | **Subtotal** | | | **28.64 ± 0.20 [a]** | | | |
| Aldehydes | ALD1 | 2-Decenal, (E)- | 3913-81-3 | Soap, tallow | n.d | n.d | 1.86 ± 0.06 [a] | 1.66 ± 0.10 [b] |
| | ALD2 | Benzaldehyde | 100-52-7 | Sweet, almond, cherry | n.d | 17.79 ± 0.09 [c] | 23.54 ± 0.05 [a] | 18.21 ± 0.05 [b] |
| | ALD3 | Benzaldehyde, 2,4-dimethyl- | 15764-16-6 | Cherry, almond, vanilla | 25.95 ± 0.04 [a] | 2.72 ± 0.05 [d] | 12.67 ± 0.06 [b] | 2.97 ± 0.09 [c] |
| | ALD5 | Benzeneacetaldehyde | 122-78-1 | Rose-like, honey, floral | 1.36 ± 0.04 [d] | 44.59 ± 0.07 [b] | 92.56 ± 0.07 [a] | 40.11 ± 0.09 [c] |
| | ALD6 | Decanal | 112-31-2 | Soap, tallow | n.d | n.d | n.d | 1.33 ± 0.08 [a] |
| | ALD7 | Heptanal | 111-71-7 | Fat, citrus, rancid | 1.12 ± 0.03 [a] | n.d | n.d | n.d |
| | ALD8 | Hexanal | 66-25-1 | Green, sweet | n.d | n.d | n.d | 2.72 ± 0.06 [a] |
| | ALD9 | Paraldehyde | 123-63-7 | Sweet, pleasant | n.d | n.d | n.d | 7.27 ± 0.07 [a] |
| | | **Subtotal** | | | **28.43 ± 0.10 [d]** | **65.10 ± 0.06 [c]** | **130.62 ± 0.03 [a]** | **74.27 ± 0.08 [b]** |
| Esters | EST1 | 1-Butanol, 2-methyl-, acetate | 624-41-9 | Fruity, floral | n.d | 16.90 ± 0.06 [b] | 16.34 ± 0.04 [c] | 27.00 ± 0.07 [a] |
| | EST2 | 1-Butanol, 3-methyl-, acetate | 123-92-2 | Fruity, floral, sweet | 0.71 ± 0.04 [d] | 59.18 ± 0.06 [b] | 54.99 ± 0.04 [c] | 91.51 ± 0.07 [a] |
| | EST3 | 1-Methoxy-2-propyl acetate | 108-65-6 | Fruity | n.d | 1.88 ± 0.09 [a] | 1.92 ± 0.05 [a] | n.d |
| | EST4 | 2-Butanol, 3-methyl-, acetate | 5343-96-4 | Fruity | n.d | n.d | 1.91 ± 0.05 [a] | n.d |

**Table 4.** *Cont.*

| Volatile Groups | SN | Compound Name | CAS Number | Odor Description | Concentration of JP, ng/100 g FW (Mean ± SD) | | | |
|---|---|---|---|---|---|---|---|---|
| | | | | | JCON | JLcAp | JLhAp | JLpAp |
| | EST5 | 3-Hexenoic acid, ethyl ester | 2396-83-0 | Fruity, brandy, wine-like | n.d | 1.43 ± 0.11 [b] | 1.91 ± 0.04 [a] | 1.27 ± 0.10 [b] |
| | EST6 | 3-Methyl-3-buten-1-ol, acetate | 5205-07-2 | Fruity | n.d | 2.39 ± 0.09 [b] | n.d | 2.82 ± 0.06 [a] |
| | EST7 | 7-Octenoic acid, ethyl ester | 35194-38-8 | Fruity | n.d | 2.66 ± 0.06 [b] | n.d | 2.84 ± 0.06 [a] |
| | EST8 | Acetic acid, 2-phenylethyl ester | 103-45-7 | Fruity, rose | n.d | 65.99 ± 0.06 [c] | 76.05 ± 0.05 [b] | 82.18 ± 0.06 [a] |
| | EST9 | Acetic acid, butyl ester | 123-86-4 | Fruity, sweet, solvent | n.d | n.d | n.d | 2.18 ± 0.03 [a] |
| | EST10 | Acetic acid, pentyl ester | 628-63-7 | Fruity, apple | n.d | n.d | n.d | 0.77 ± 0.10 [a] |
| | EST11 | Acetic acid, phenylmethyl ester | 140-11-4 | Fruity | n.d | 4.99 ± 0.08 [b] | 4.98 ± 0.05 [b] | 7.03 ± 0.05 [a] |
| | EST12 | Acetoin acetate | 4906-24-5 | Fruity | n.d | 7.82 ± 0.08 [a] | n.d | n.d |
| | EST13 | Benzeneacetic acid, ethyl ester | 101-97-3 | Fruity | n.d | n.d | n.d | 1.45 ± 0.03 [a] |
| | EST14 | Benzenepropanoic acid, ethyl ester | 2021-28-5 | Fruity | n.d | 16.82 ± 0.09 [a] | n.d | 16.17 ± 0.06 [b] |
| | EST15 | Benzoic acid, ethyl ester | 93-89-0 | Fruity, floral | 2.64 ± 0.03 [d] | 6.64 ± 0.04 [c] | 10.97 ± 0.10 [a] | 8.00 ± 0.06 [b] |
| | EST16 | cis-9-Tetradecenoic acid, propyl ester | | Fruity | n.d | 1.25 ± 0.10 [a] | 0.98 ± 0.07 [b] | 0.86 ± 0.07 [b] |
| | EST17 | Dodecanoic acid, ethyl ester | 106-33-2 | Fruity | n.d | 5.56 ± 0.03 [b] | 6.33 ± 0.08 [a] | 4.04 ± 0.04 [c] |
| | EST18 | Dodecyl tiglate | 1231959-17-3 | Fruity | n.d | n.d | n.d | 1.05 ± 0.03 [a] |
| | EST19 | Ethyl 4-acetoxybutanoate | 25560-91-2 | Fruity | n.d | 0.93 ± 0.04 [b] | n.d | 1.05 ± 0.03 [a] |
| | EST20 | Ethyl 9-hexadecenoate | 54546-22-4 | Fruity | n.d | 0.91 ± 0.09 [a] | n.d | n.d |
| | EST21 | Ethyl Acetate | 141-78-6 | Pineapple, fruity | n.d | n.d | 0.99 ± 0.03 [a] | 0.80 ± 0.06 [b] |
| | EST22 | Heptanoic acid, ethyl ester | 106-30-9 | Wine-like, fruity, brandy | n.d | n.d | 1.58 ± 0.09 [a] | n.d |
| | EST23 | Hexadecanoic acid, ethyl ester | 628-97-7 | Fruity | n.d | 2.87 ± 0.04 [a] | 1.92 ± 0.06 [b] | 1.37 ± 0.07 [c] |
| | EST24 | Hexanoic acid, ethyl ester | 123-66-0 | Fruity, apple, banana | 16.70 ± 0.03 [b] | 8.24 ± 0.09 [d] | 20.41 ± 0.06 [a] | 9.75 ± 0.06 [c] |

**Table 4.** *Cont.*

| Volatile Groups | SN | Compound Name | CAS Number | Odor Description | Concentration of JP, ng/100 g FW (Mean ± SD) | | | |
|---|---|---|---|---|---|---|---|---|
| | | | | | JCON | JLcAp | JLhAp | JLpAp |
| | EST25 | Isoamyl lactate | 19329-89-6 | Fruity, banana, pear | n.d | 1.31 ± 0.10 [a] | n.d | 0.79 ± 0.11 [b] |
| | EST26 | Methyl salicylate | 119-36-8 | Fruity, peppermint | 2.69 ± 0.02 [a] | n.d | n.d | n.d |
| | EST27 | Octanoic acid, ethyl ester | 106-32-1 | Brandy, pear, musty | n.d | 3.01 ± 0.09 [b] | 5.31 ± 0.03 [a] | 2.92 ± 0.03 [b] |
| | EST28 | Pentanoic acid, ethyl ester | 539-82-2 | Fruity, berry | n.d | 2.17 ± 0.03 [c] | 2.28 ± 0.06 [b] | 2.43 ± 0.06 [a] |
| | EST29 | Propanoic acid, 2-hydroxy-, ethyl ester | 97-64-3 | Fruity | n.d | 2.95 ± 0.10 [a] | n.d | 0.96 ± 0.06 [b] |
| | EST30 | Propanoic acid, 2-methyl-, 3-hydroxy-2,2,4-trimethylpentyl ester | 77-68-9 | Fruity | n.d | 1.67 ± 0.10 [b] | 1.86 ± 0.07 [a] | 1.98 ± 0.05 [a] |
| | EST31 | Tetradecanoic acid, ethyl ester | 124-6-1 | Soap, mild, polish | n.d | 1.35 ± 0.07 [a] | n.d | n.d |
| | | **Subtotal** | | | **22.75 ± 0.09 [d]** | **220.86 ± 0.67 [b]** | **210.71 ± 0.24 [c]** | **271.22 ± 0.68 [a]** |
| Other(s) | NAT1 | Naphthalene, 1,2,3,4-tetrahydro-1,1,6-trimethyl- | 475-3-6 | tar | 7.57 ± 0.03 [a] | n.d | n.d | n.d |
| | | **Total of volatile compounds** | | | **611.07 ± 3.60 [a]** | **522.38 ± 0.77 [d]** | **599.85 ± 0.33 [b]** | **568.49 ± 0.55 [c]** |

Data expressed as mean ± standard deviation. Means in the same row with different superscript letters are significantly different ($p < 0.05$). **Note:** CAS—chemical abstracts service; n.d—not detected; JLcAp—*Lacticaseibacillus casei* Lc 122—*Acetobacter pasteurianus* Ap-As.1.41 HuNiang 1.01 puree; JLhAp—*Lactobacillus helveticus* Lh 43—*A. pasteurianus* Ap-As.1.41 HuNiang 1.01 puree; JLpAp—*Lactiplantibacillus plantarum*—*A. pasteurianus* Ap-As.1.41 HuNiang 1.01 puree; (JCON)—puree with no bacteria inoculants (unfermented).

### 3.7. Volatile Profiling Using HS-SPME/GC–MS

Our investigation employed qualitative and quantitative analyses using headspace solid-phase microextraction (HS-SPME) coupled with gas chromatography–mass spectrometry (GC–MS) to evaluate the volatiles demonstrated using the electronic nose technique comprehensively. The findings presented in Tables S2 and 4 encompassed the identification of 80 volatile compounds with distinct aroma profiles in bicultured samples (JLcAp, JLhAp, JLpAp) and the control (JCON). Quantitatively, the compounds exhibited concentrations of 522.38, 599.85, 568.49, and 611.07 ng/100 g in JLcAp, JLhAp, JLpAp, and JCON, respectively. Esters predominated, followed by acids, alcohols, and aldehydes. The significantly higher total volatile concentration in the control sample could be attributed to the combined effect of alcohol and acids, suggesting inefficient bioconversion of these volatiles in the unfermented sample, resulting in residues. This also implicated the most ester formation in the bicultured samples largely due to the use of these acids and alcohols for esterification by lactic and acetic acid bacteria activities. The results showed significant agreement with previous research by Li et al. [17] and Liu et al. [30]. Maximum concentrations of esters, acids, alcohols, and aldehydes were evident in JLcAp (522.40 ng/100 g), JLhAp (599.84 ng/100 g), JLpAp (568.50 ng/100 g), and JCON (574.87 ng/100 g). Benzeneacetaldehyde emerged as the most abundant aldehyde, particularly in JLhAp (92.56 ng/100 g), exhibiting a decreasing trend in JLcAp and JLpAp, with significant differences ($p < 0.05$), reducing in JCON to 1.36 ng/100 g. A contrasting pattern was observed for 2,4-dimethyl benzaldehyde, with concentrations of 0.52%, 2.11%, and 0.52% in JLcAp, JLhAp, and JLpAp, respectively, from their initial value of 4.25% (JCON). The synergistic interactions among bicultures led to the production of specific aldehydes, such as benzaldehyde, which are absent in JCON. Moreover, 2,4-dimethyl-benzaldehyde and benzeneacetaldehyde were synthesized by all bicultures, potentially contributing to desirable fruity sensory notes [46]. The observed bioconversion of aldehydes was attributed to the auto-oxidation of unsaturated fatty acids, as suggested by Song et al. [15]. In the unfermented sample, JCON, seventeen volatile acid compounds were identified. However, the bicultured Junzao jujube purees exhibited a reduced concentration, with 7 to 10 acids identified. Isobutyric acid and butanoic acid were initially reported by Chen et al. [47] in *Ziziphus jujuba* cv. Junzao were exclusively detected during the red maturity stage. Dodecanoic, heptanoic, hydrocinnamic, n-decanoic, and octanoic acid consistently emerged in the bicultured and unfermented samples. Among the nineteen acids profiled, 2-octenoic and 3-decenoic acids were remarkably absent in JCON and manifested after the lactic–acetic acid fermentation in all bicultures of JLcAp and JLhAp only. The decline in acid content in the unfermented sample provided insights into potential alcohol and acid utilization during processes like esterification, contributing to the formation of flavoring compounds. A notable disparity was observed in esters, with 17 to 24 identified in the bicultured samples compared to 4 in JCON. On average, this represented a significant 42% increase in ester formation in the fermented samples, with concentrations of 220.86 ng/100 g (JLcAp), 210.71 ng/100 g (JLhAp), 271.22 ng/100 g (JLpAp), and 22.75 ng/100 g (JCON). The diversity in ester volatiles highlighted the distinctive contributions of specific strains in synergistic interactions, resulting in the formation of uncommon volatiles. This diversity may account for the heightened fruity, floral, and delightful characteristics of bicultured jujube purees compared to JCON. Total alcohol content exhibited the highest concentration in JLpAp (31.37%), featuring 11 alcohol compounds, whereas JCON, JLcAp, and JLhAp, featured 10 alcohol compounds at 333.20 ng/100 g (54.53%), 177.43 ng/100 g, and 176.61 (29.44%) respectively. Among bicultured samples, six alcohols (1-hexanol, 2-ethyl-hexanol, 2,3-butanediol, 2,4-di-tert-butylphenol, benzyl alcohol, phenylethyl alcohol) were consistently identified. The data suggested that 2,3-butanediol was pivotal as the primary alcohol in Junzao jujube puree, constituting nearly 43% of the total volatile concentration in the unfermented puree. However, this concentration exhibited a decline after fermentation. Conversely, 1-decanol, 1-nonanol, and 1-octanol, present in JCON, were conspicuously absent in all bicultured purees. This decline may be attributed to the possible esterification of these alcohols with available acids in the unfermented sam-

ple, contributing to the production of highly aromatic fermented products. Concurrently, certain volatiles—such as 1,2,3,4-tetrahydro-1,1,6-trimethyl (naphthalene)—experienced a reduction and remained undetected in the fermented samples.

### 3.8. Effect of Bicultural Fermentation on the Microstructural Changes

Scanning electron microscopy (SEM) analysis was performed to assess the impact of co-fermentation on the structure of jujube puree. As illustrated in Figure 2d, the surface of unfermented jujube puree (JCON) displayed a slightly curled, sheet-like structure, with most maintaining their integrity. The curled, sheet-like structure might result from the high sucrose content [1] in dry jujube fruit. Following fermentation, the surface of the unfermented jujube puree (JCON) stretched, became coarse, and exhibited major alteration microscopically being modified as minor holes, indicating the breakdown of the rigid, curled, sheet-like structure during the lactic–acetic co-fermentation process. Additionally, carbohydrate hydrolases in lactic and acetic acid bacteria could more readily interact with the disintegrated jujube puree samples, resulting in apparent elongated "sheets" on the surface in fermented samples. This occurrence could be attributed to the undigested nature of dietary fiber in jujube puree, acting as a protective barrier for phenolic compounds. These microstructural changes might be largely associated with the formation of micelles in the fermented samples [30,37], decreasing the availability of phenolic compounds.

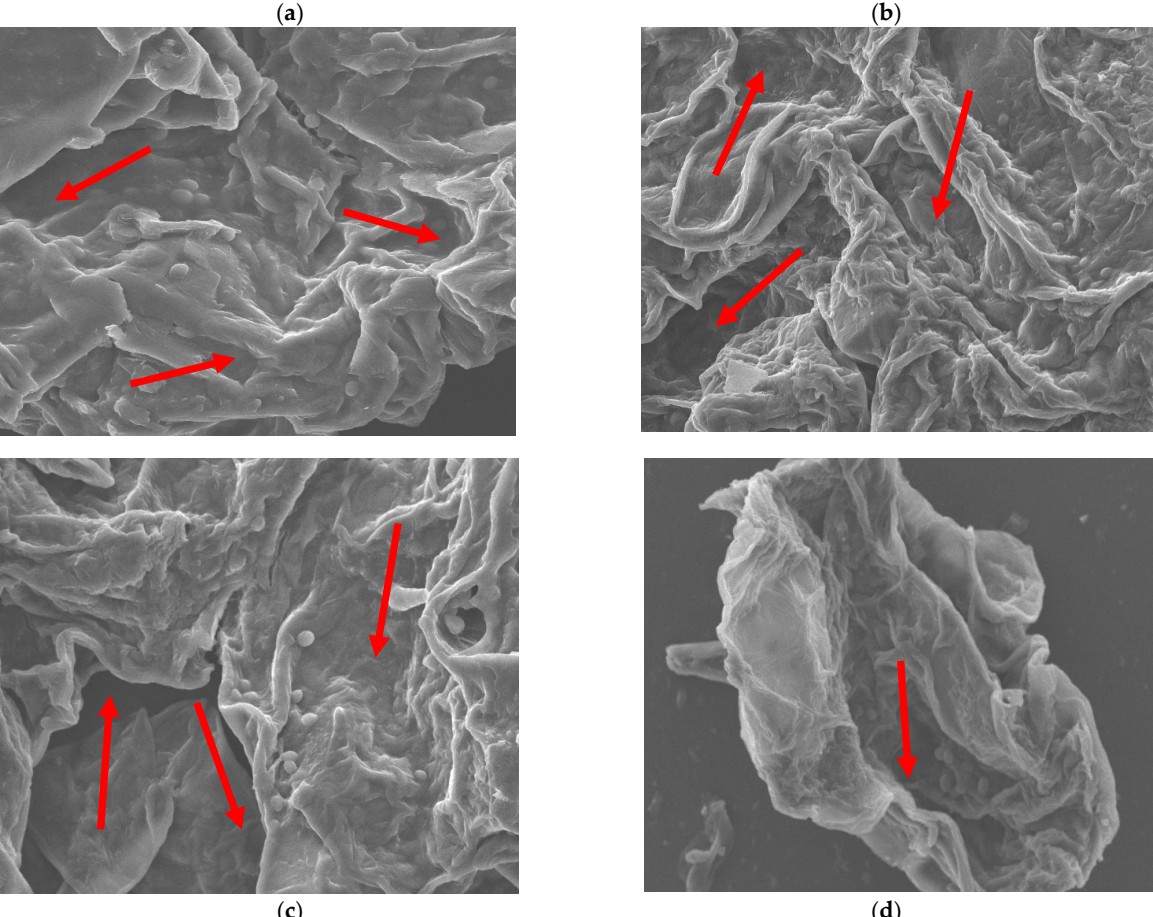

**Figure 2.** Scanning electron micrographs (×1000, 10 kV) of bicultured jujube purees. (**a**)—JLcAp; (**b**)-JLhAp; (**c**)—JLpAp; (**d**)—control. **Note:** JLcAp—*Lacticaseibacillus casei* Lc 122—*Acetobacter pasteurianus* Ap-As.1.41 HuNiang 1.01 puree; JLhAp—*Lactobacillus helveticus* Lh 43—*A. pasteurianus* Ap-As.1.41 HuNiang 1.01 puree; JLpAp—*Lactiplantibacillus plantarum*—*A. pasteurianus* Ap-As.1.41 HuNiang 1.01 puree; (JCON)—puree with no bacteria inoculants (unfermented).

### 3.9. Principal Component Analysis (PCA)

PCA, a potent tool for dimensionality reduction, serves the technical purpose of revealing distinctive correlation patterns [48] and preserving a substantial portion of data to evaluate the overall influence of diverse treatments on total phenolic contents, total flavonol content, anthocyanin contents, volatile groups, and free amino acid profiles. Additionally, it aims to identify potential clusters among bicultures and the unfermented sample. The output (Figure 3) demonstrated that PCs 1, 2, and 3 collectively accounted for an impressive 97.73% of the total variance, affirming the robustness of the applied methods [18,38]. The samples are categorized into four distinct components, each associated with specific attributes. Sample JLcAp exhibited close associations with elevated anthocyanin contents and a moderate amount of ester groups. The control displayed high volatiles: ketones, alcohols, and free amino acids: proline, aspartic acid, alanine, glycine, isoleucine, and arginine. Sample JLhAp was heavily clustered with most attributes—predominantly free amino acids: phenylalanine, tyrosine, valine, threonine, leucine, lysine, serine glutamic acid—and relatively characterized by a total phenolic content and total flavonol content. Puree JLpAp showcases elevated levels of methionine, predisposing it to have an advantage of full essential amino acid content at the expense of the other bicultures (JLcAp and JLhAp) and unfermented sample (JCON). In summary, principal component analysis (PCA) elucidated that the lactic–acetic acid co-fermented jujube purees significantly impacted various functional indices. This effect is attributed to the distinct strains that noticeably influenced the phenolic contents, free amino acids, and volatile groups of the bicultured purees, microscopically modified by SEM.

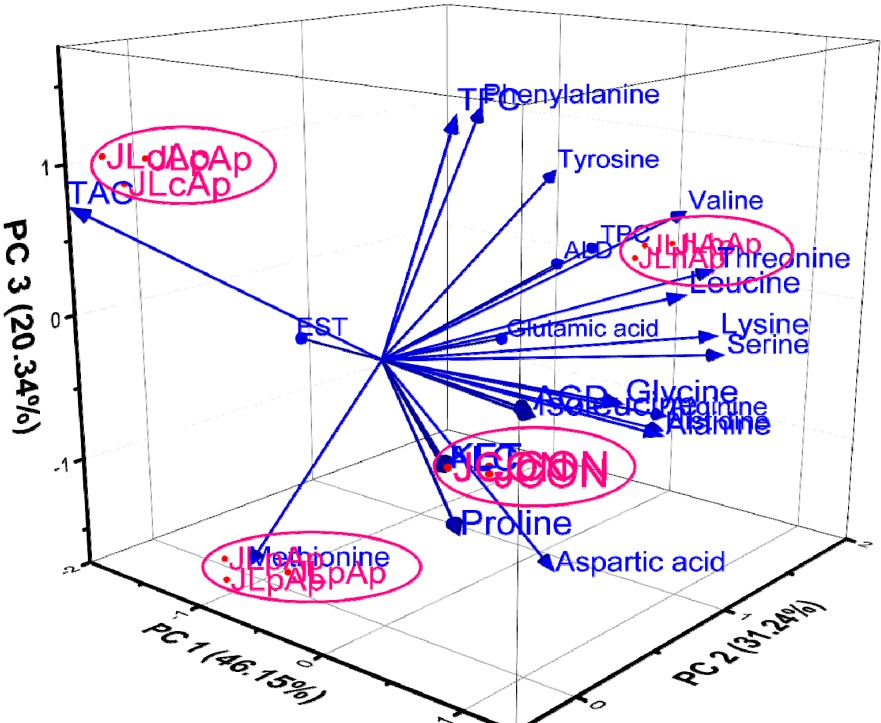

**Figure 3.** Three-dimensional loading plot for PC1-PC2-PC3 of bicultured Junzao jujube purees and control showing linkages between variables: free amino acids, phenolic profile groups, and volatile groups. **Note:** TPC—total phenolic acid content; TFC—total flavonol content; TAC—total anthocyanin content; ALC—alcohol group; ACD—acid group; EST—ester groups; ALD—aldehyde group; KET—ketone groups. JLcAp—*Lacticaseibacillus casei* Lc 122—*Acetobacter pasteurianus* Ap-As.1.41 HuNiang 1.01 puree; JLhAp—*Lactobacillus helveticus* Lh 43—*A. pasteurianus* Ap-As.1.41 HuNiang 1.01 puree; JLpAp—*Lactiplantibacillus plantarum*—*A. pasteurianus* Ap-As.1.41 HuNiang 1.01 puree; (JCON)—puree with no bacteria inoculants (unfermented).

## 4. Conclusions and Future Studies

Bicultured purees demonstrated significant probiotic effects, exhibiting viable counts exceeding 6–7 log CFU/mL. The free amino acid (FAA) composition of bicultured Junzao jujube purees exhibited variability, with JLhAp displaying the highest content, followed by intermediate levels in JLpAp and the lowest in JLcAp. Abundant FAAs included L-aspartic acid and proline, with the noteworthy observation of methionine synthesis in JLpAp. This emphasizes dynamic enzymatic transformations facilitated by the interplay of *Lactiplantibacillus plantarum* Lp 28 and *Acetobacter pasteurianus* Ap-As.1.41 HuNiang 1.01 biculture. Furthermore, a notable correlation emerged between phenolic compounds (specifically chlorogenic acid, rutin, and *p*-coumaric acid) and antioxidant capacities, assessed through ABTS-RSA and DPPH-RSA measurements. A significant negative correlation was also observed between overall color difference and cyanidin 3-O-rutinoside, emphasizing the influence of this compound on color variations. The analysis identified 80 volatile compounds in the lactic–acetified jujube purees, with esters emerging as the most abundant class. Ten to eleven alcohols were identified, with 2,3-butanediol prevailing as the primary alcohol in Junzao jujube puree. Interestingly, its concentration decreased post-fermentation, potentially due to esterification with unfermented acids. SEM analysis revealed distinct structural changes, with unfermented jujube puree (JCON) displaying a curled, sheet-like structure, possibly attributed to undigested dietary fiber and high sucrose content in dry jujube fruit. *Lp. plantarum* Lp 28 and *A. pasteurianus* Ap-As.1.41, HuNiang 1.01 showcased significantly higher amounts of essential amino acids, positioning them as potentially optimal formulations. These findings highlight the impactful modifications resulting from lactic–acetic acid co-fermentation on the evaluated quality indices. The study advocates for further exploration through detailed metabolomic investigations to unveil the intricate biochemical processes within these strains. Such an approach holds promise for advancing our understanding of biological systems, health, and disease, with potential applications in scientific and practical domains.

**Supplementary Materials:** The following supporting information can be downloaded at: https://www.mdpi.com/article/10.3390/fermentation10010071/s1.

**Author Contributions:** T.A.B.: conceptualization, methodology, writing—original draft, writing—review and editing, data analysis, visualization. Y.X.: methodology, writing—review and editing. I.D.B.: review of information, data analysis, writing—review and editing. J.A.: writing—review and editing. Y.L.: writing—review and editing. K.C.: writing—review and editing. A.Y.A.: methodology, writing—review and editing. S.Y.: writing—review and editing. Y.M.: conceptualization, resources, supervision, project administration, funding acquisition, review of information, draft approval, and final manuscript approval. All authors have read and agreed to the published version of the manuscript.

**Funding:** The authors are grateful to the management of Zhenjiang Key Research and Development Program (Modern Agriculture) for their support (grant number NY2020020).

**Institutional Review Board Statement:** Not applicable.

**Informed Consent Statement:** Not applicable.

**Data Availability Statement:** Data is available upon reasonable request from the corresponding author.

**Conflicts of Interest:** The authors declare no conflicts of interest.

## Abbreviations

| | |
|---|---|
| AAB | acetic acid bacteria |
| ABTS-RSA | 2,2-azino-bis-3-ethylbenzothiazoline-6-sulfonic acid radical scavenging activity |
| *b** | yellowness-blueness |
| DPPH-RSA | 2,2-diphenyl-1-picrylhydrazyl radical scavenging activity |
| E-nose | electronic nose |
| FAA | free amino acid |
| HPLC | high-performance liquid chromatography |
| HS-SPME/ | |
| GC–MS | headspace solid-phase microextraction gas chromatography–mass spectrometry |
| JCON | pasteurized puree without inoculation |
| JLcAp | *Lacticaseibacillus casei* Lc 122-*Acetobacter pasteurianus* Ap-As.1.41 HuNiang 1.01 puree |
| JLhAp | *Lactobacillus helveticus* Lh 43-*A. pasteurianus* Ap-As.1.41 HuNiang 1.01 puree |
| JLpAp | *Lactiplantibacillus plantarum* Lp 28-*A. pasteurianus* Ap-As.1.41 HuNiang 1.01 puree |
| JP | bicultured jujube puree |
| LAB | lactic acid bacteria |
| SEM | scanning electron microscopy |
| TAC | total anthocyanin content |
| TFC | total flavonol content |
| TPC | total phenolic content |

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
