# Peer review of "Innovative Bicultured Lactic–Acetic Acid Co-fermentation Improves Jujube Puree’s Functionality and Volatile Compounds"

_fermentation, doi:10.3390/fermentation10010071_

Round 1

Reviewer 1 Report

Comments and Suggestions for Authors

The article titled ‘Innovative bi-cultured lactic-acetic acid co-fermentation improves jujube puree’s functionality and volatile compounds.’ has been reviewed. Some changes are needed before publishing the manuscript in Fermentation journal. The article is well-written and planned. However, I hardly recommend analyzing some bioactivities…such as antioxidant (DDPH, ABST, cellular) due to phenolic compounds content. If not, the article is just descriptive.

Line 33 You can short the bacteria name since you wrote the species and genus previously. Example: L. plantarum

Line 37 SEM instead of the explanation since you mentioned in line 27 the meaning of the abbreviation.

Line 42 Is the word bacteria missing in the keyword ‘lactic and acetic acid’?

Line 73 extra space

Line 144 Include the abbreviation LAB and AAB every time you refer to these bacteria. Check throughout the manuscript.

Line 136 How did you check the color? Did you use a color scale or colorimeter? Please add

Line 237 Did you previously checked if there was 2-octanol content in the jujube puree product?

Line 242 Please add the exposure time after the equilibration time (20 min).

Line 311 Again the LAB and AAB abbreviation are explained, please only described the meaning once.

Line 335 Different letter type, unify all the manuscript.

Line 363 Different reference type, unify all.

Line 469 Please add the CAS numbers in Table 3. Could you compare the VOCs PCA vs the E-nose PCA? Then maybe both techniques can detect same differences and only few market compounds are interesting.

Line 493 You might include the aromatic properties of some of the main compounds, to describe aromatic properties of the product.

Line 587 At the end, which combination was the best? Maybe you could give an aromatic profile of the purees depending on the bacteria added. Same with the phenolic compounds and their bioactivity.

Author Response

Attached is the response to the comments.

Reviewer 2 Report

Comments and Suggestions for Authors

The manuscript describes an interesting study aimed at demonstrating that a combined lactic-acetic fermentation is useful to revalue Junzao jujube puree, improving its organoleptic characteristics and extending the shelf life. The study is aimed above all at evaluating the fermented product from a chemical and sensory point of view, with good results and adequate methodology in this aspect. The design is adequate, the objectives are clear, the English is good but it has neglected to describe the microbiological changes during fermentation. The observations are detailed below.

- the genus and species of the microorganisms must always be written in italics. Check References

-When talking about the strains used, genus, species and identification must necessarily be indicated. If I only write genus and species, it can be interpreted that all the strains of that genus and species give the same results, which is not the case (strain dependence). See Abstract, point 2.3, etc.

-line 173: "lactobacilli" or "Lactobacillus"

-In my opinion, the main methodological problem of the study is not showing the evolution of the counts of the strains used during fermentation. If this is not done, it is impossible to relate chemical and sensory changes to the microbiology of the product. Selective counts of lactobacilli and acetic bacteria will show: initial count and its eventual growth to a final count. This evolution is responsible for the final characteristics of the product. It could be that a change is attributed to a microbial group that, looking at its counts, has not even grown in the fermentation. The description of a fermentation must necessarily include microbiological, chemical and sensory changes.

​

Author Response

(The authors gave the same response as above.)

Round 2

Reviewer 1 Report

Comments and Suggestions for Authors

The authors carried out the changes I proposed. I think they highly improved the masnucript including the antioxidant and color analysis. 

Only few minor changes:

The CAS number of EST6 is with / instead of -. Please check. 

Whay do you mean CAS 0-0-0 in Table 4? Add the meaning somewhere or remove the number. 

Author Response

find attached the response to comments.

Reviewer 2 Report

Comments and Suggestions for Authors

Some recommendations were partially fulfilled. It is detailed:

-the genus and species of the microorganisms must always be written in italics. See lines 692, 763, 777/8...Review all text

- the identification of the strains must be written whenever they appear in the text. See 2.3, line 178. Review all text

The manuscript appears improved in its latest version

Author Response

find attached the response to comments
